# Precision Medicine for Nasopharyngeal Cancer—A Review of Current Prognostic Strategies

**DOI:** 10.3390/cancers16050918

**Published:** 2024-02-24

**Authors:** Luvita Suryani, Hazel P. Y. Lee, Wei Keat Teo, Zhi Kang Chin, Kwok Seng Loh, Joshua K. Tay

**Affiliations:** 1Department of Otolaryngology-Head & Neck Surgery, Yong Loo Lin School of Medicine, National University of Singapore, Singapore 119228, Singapore; luvita.s@nus.edu.sg (L.S.); enttwk@nus.edu.sg (W.K.T.); zkchin@nus.edu.sg (Z.K.C.);; 2Department of Otolaryngology-Head & Neck Surgery, National University Hospital, Singapore 119228, Singapore; peiyu.lee@mohh.com.sg

**Keywords:** nasopharyngeal carcinoma, Epstein–Barr virus, risk stratification, precision medicine, radiomics, genomic, biomarker

## Abstract

**Simple Summary:**

The current standard of care for nasopharyngeal carcinoma (NPC) is predominantly guided by the American Joint Committee on Cancer (AJCC)/Union for International Cancer Control (UICC) tumour-node-metastasis (TNM) staging system. This review aims to discuss the current strategies and development of new technologies to optimise NPC patients’ stratification to ultimately provide better patient management and more targeted therapy.

**Abstract:**

Nasopharyngeal carcinoma (NPC) is an Epstein–Barr virus (EBV) driven malignancy arising from the nasopharyngeal epithelium. Current treatment strategies depend on the clinical stage of the disease, including the extent of the primary tumour, the extent of nodal disease, and the presence of distant metastasis. With the close association of EBV infection with NPC development, EBV biomarkers have shown promise in predicting treatment outcomes. Among the omic technologies, RNA and miRNA signatures have been widely studied, showing promising results in the research setting to predict treatment response. The transformation of radiology images into measurable features has facilitated the use of radiomics to generate predictive models for better prognostication and treatment selection. Nonetheless, much of this work remains in the research realm, and challenges remain in clinical implementation.

## 1. Introduction

Nasopharyngeal carcinoma (NPC) is the most common malignancy arising from the nasopharyngeal epithelium [1,2]. There are approximately 129,000 new cases of NPC reported globally annually, and the incidence is predicted to continue to rise between 2020 and 2049 in China [3,4]. NPC has an extremely skewed racial and geographical distribution [5]. NPC is considered rare among Caucasians yet endemic to natives in several regions including southern China, Southeast Asia, the Arctic, the Middle East and North Africa [1,6]. The racial distribution skew is also observed in Singapore, as the annual NPC incidence in the Chinese and Malay population is significantly higher than in the Indian population [7]. NPC remains in the top ten causes of cancer deaths for males in Singapore [7]. 

NPC is radiosensitive, yet 15–58% of the patients still experience recurrence, and most will develop distant metastases [8]. These patients generally have poor prognosis with only 20 months median overall survival (OS) [9]. Current NPC stratification methods are based on the American Joint Committee on Cancer (AJCC)/Union for International Cancer Control (UICC) tumour-node-metastasis (TNM) staging system [10], and the current standard of care is based on this staging method. 

Therefore, an accurate risk stratification upon diagnosis will be crucial for an effective and efficient treatment plan. This review highlights the current efforts to improve the prognostic value of NPC risk stratification through various technologies to provide better patient management and more targeted therapy. 

## 2. Current Diagnosis, Staging, and Treatment Options

### 2.1. Diagnosis and Staging

NPC often arises from the pharyngeal recess, also known as the fossa of Rosenmüller, an anatomic area that requires an endoscopic examination for assessment. Patients are often asymptomatic during the early course of the disease. Symptoms presented often correspond to an advanced stage of disease, including a painless neck lump, blood-stained nasal discharge, nasal obstruction, hearing impairment, and cranial nerve palsies [11,12].

The diagnosis of NPC is made through the identification of EBV in tissue samples endoscopically obtained from the nasopharynx. In situ hybridisation (ISH) is used to detect EBV-encoded RNA (EBER) in the samples, and the histology of the sample is also determined.

The prognostic significance of the latest eighth edition AJCC/UICC TNM staging system has been validated in several papers, and it has provided a more accurate prediction of treatment outcomes [13,14] and a better discrimination in prognosis using the N category [15,16]. 

In this staging system, which is predominantly based on anatomic features of the tumour, imaging is crucial in determining prognosis. The recommended scans for staging, which are performed in the authors’ institutions, would be magnetic resonance imaging (MRI) of the nasopharynx and surrounding structures to characterise the extent of spread, as well as a positron emission tomography-computed tomography (PET-CT) scan to determine the presence of distant metastasis [17]. Patients are categorised into different stages corresponding to different survival outcomes to guide treatment strategies.

### 2.2. Current Treatment Options

Multiple international guidelines [17,18,19] are available regarding the treatment of NPC. Endemic EBV-positive non-keratinising undifferentiated NPC is both radiosensitive and chemosensitive; thus, they have formed the cornerstone of treatment for NPC patients. Intensity-modulated radiation therapy (IMRT) is now the mainstay treatment method, allowing accurate delivery of high doses of radiation while minimising collateral damage, improving local-recurrence-free survival rates and phasing out two-dimensional radiotherapy (RT) [20,21,22]. 

Patients with early-stage NPC (i.e., Stage I) receive RT alone, and those with Stage II NPC and above receive concurrent chemoradiotherapy (CCRT). Patients with advanced-stage NPC will be advised for either induction or adjuvant chemotherapy. Palliative chemotherapy with cisplatin and gemcitabine is recommended for recurrent and metastatic disease [23], and for locoregional recurrences, the options of surgical resection and re-irradiation remain. Molecular targeted therapy and immunotherapy are options for NPC demonstrating chemoresistance, targeting numerous molecules implicated in the carcinogenesis pathway for NPC [24]. 

There has been increasing focus on molecular targeted therapy and immunotherapy as options for NPC demonstrating chemoresistance, targeting numerous molecules implicated in the carcinogenesis pathway for NPC [24]. With the strong association of EBV with the pathogenesis of NPC, there has been ongoing research on therapeutic EBV targets. The tumour microenvironment of EBV-driven NPC is favourable for immunotherapy. NPC expresses a number of EBV antigens, including LMP1, LMP2A/B, EBNA1, EBER, and EBV-encoded RNA, which contributes to tumourigenesis, and a better understanding of this process will help in the development of EBV-targeted immunotherapies [25]. 

Therapeutic vaccines for EBV that target EBV antigens have been in the works, using autologous dendritic cells to trigger T-cell responses specific to EBV antigens, although the clinical response seen in the various trials has not been consistent [26]. EBV-specific cytotoxic T lymphocytes are also a potential treatment option, with a phase II study demonstrating it to have a significantly higher overall survival and a favourable safety profile [25]. Apart from EBV targets, immune checkpoint inhibitors have been used for the treatment of recurrent and/or metastatic NPC. Anti-PD-1 antibodies have shown positive outcomes, such as the addition of camrelizumab to the standard gemcitabine and cisplatin chemotherapeutic agents in the CAPTAIN-1st trial, which increased progression-free survival significantly in patients with recurrent or metastatic NPC [26]. The biomarkers associated with NPC, some of which will be touched on in this paper, are unique due to their relations to EBV. More research is still required on how epigenetic changes such as DNA methylation, microRNA expression, and cell-free EBV DNA play a role in the selection of patients and prediction of immunotherapy response [27,28].

## 3. Limitations of the Current Staging System in Predicting Prognosis

The AJCC/UJCC TNM NPC staging system is updated at least once every decade. The eighth edition saw the addition of EBV titres, as well as the T0 category for patients with metastatic lymph node(s) of unknown primary tumours that are EBV-positive. However, there are still limitations. It misses out on the clinical, molecular, and histopathological factors that are involved in the pathogenesis of NPC, and it has been shown that patients grouped under the T2 and T3 stages have similar survival [29]. It is also deficient in risk-stratifying patients with de novo metastatic NPC (dmNPC) [30,31]. A new surgical staging system, based on the current recurrent tumour, node, and metastasis (rTNM) staging system, as well as imaging data, has been proposed, improving the prognostic value of the existing rTNM system, with the surgical stage being a significant prognostic factor for overall survival [32].

With better risk stratification and staging, high-risk patients can be appropriately selected to undergo more aggressive treatment, with improved outcomes [33]. This is particularly important as the more aggressive treatment will likely have increased toxicity, including hearing impairment, osteoradionecrosis of the external auditory canal and skull base, cranial perve palsies, to devastating carotid blowouts [34,35,36]. There have been positive results from adding induction chemotherapy to concurrent CCRT for locoregionally advanced NPC [37,38,39], but when taking into account the drawbacks, including treatment complications, increased duration, and thus the cost of treatment, the marginal benefits of induction chemotherapy may not be worth it [40]. Hence, more research is needed to elicit factors that can identify patients who will benefit from more intensive therapy without subjecting them to unnecessary complications. There has been a growing interest in combining non-anatomical prognostic factors, including clinical and molecular biomarkers, to the traditional TNM staging [41] to improve risk stratification, which will be discussed in the following paragraphs.

## 4. Clinical Biomarkers for Risk Stratification

### 4.1. Radiomics

Radiomics involves the extraction of image-based features, known as radiomic features, from medical images, transforming tumour shape, texture, intensity and wavelet features into analysable data, which may otherwise not be captured with conventional analysis. Radiomic features can be classified into semantic and agnostic features. Semantic features could be either qualitative or quantitative and include shape, size, location, vascularity, and spiculation, which can be used to identify and characterise disease, while agnostic features are further classified into first, second, and higher-order features and allow the quantification of tumour heterogeneity based on image intensity [42]. Patients with NPC frequently undergo scans for initial staging as well as for disease surveillance post-treatment. The imaging data could be used to generate predictive models to stratify patients upon diagnosis and after treatment for prognostication, selection of therapy and better outcomes. 

In a retrospective study by Ouyang et al. [43], a radiomics signature was constructed using five radiomic features correlating most to progression-free survival (PFS) out of the 970 that were extracted from pre-treatment MR images of patients with advanced-stage NPC. A Rad-score was derived from this signature, which could be used to stratify patients into low and high risk for PFS. This model was successfully validated within the study with a separate group of 30 patients, performing better than other clinical biomarkers such as age, gender, overall stage and haemoglobin and platelet counts. 

Another multicentre retrospective study [44] identified pre-treatment MRI radiomic features associated with PFS that can predict 3-year disease progression after primary treatment in non-metastatic NPC patients. One of the features was low tumour sphericity, which generally reflects a higher risk of early disease progression. However, this study also observed that low sphericity indicated a lower risk of stage I and II tumours than stage II and IV tumours. This was postulated to be because early-stage tumours tend to follow the shape of the nasopharynx, which has an irregular shape and thus might explain the low sphericity value. Another feature identified was a high mean absolute deviation value in contrast-enhanced T1-weighted scans, which reflects greater intratumoural distribution of intensities, and, thus, a more heterogenous nature of the tumour. Patients with a high risk for poor outcomes could be identified with these data and be given more aggressive treatment.

Radiomic features could be extracted from F-18 fluorodeoxyglucose (^18^F-FDG) PET scans as well to quantify tumour heterogeneity and have been shown to be superior to conventional PET parameters, such as total lesion glycolysis (TLG), in prognosticating treatment outcomes [45]. Uniformity (*p* = 0.001), which quantifies image homogeneity and, thus, tumour homogeneity, independently predicted OS, and a higher histogram-based factor skewness independently predicted a lower recurrence-free survival (RFS) (*p* = 0.008). By combining with the clinical factors of EBV DNA load and age, the prognostic value of radiomic features significantly improved.

A similar study by Intarak et al. extracted radiomic features from CT scans of non-Chinese patients who were newly diagnosed with NPC. These features, representing the size and shape, homogeneity, and similarity of the grey-level patterns of the region of interest, together with clinical features such as age, T-stage, and pre-treatment EBV DNA level, were selected and compared. Based on the highest area under the receiver operating characteristic (ROC) curve (AUC) and Harrell’s C-index of overall survival, progression-free survival, and distant metastasis-free survival (DMFS), CT-based radiomic features helped to predict survival outcomes when used together with clinical features rather than when used individually [46].

Radiomic features of different scan modalities have also been utilised concurrently for risk stratification. A radiomics nomogram constructed from 18 CT-based and PET-based features associated with disease-free survival (*p* < 0.001) successfully stratified patients into low-risk and high-risk groups for OS, DMFS and locoregional relapse-free survival (LRRFS) [47]. This could allow for individualised treatment strategies where the high-risk patients would be more suitable for induction chemotherapy. This nomogram also showed a higher C-index and, thus, a stronger prognostic value for DFS than models based on clinical factors and EBV DNA levels. Another study also found that the usage of both PET and/or CT radiomics features with clinical parameters had a similar or better prognosis through the comparison of C-index values than the usage of only PET or CT or clinical parameters alone [48]. 

Lastly, radiomics can contribute to treatment regimes by identifying specific areas of the tumour that have a higher chance of recurrence, as well as assessing tumour responses during ongoing RT [49] and post-treatment [50] for more targeted therapy towards high-risk sites. A retrospective multicohort study has found that for patients with advanced solid tumours, including NPC, radiomics could also be used to assess CD8 cell tumour infiltration and elicit tumour immune phenotypes that can predict tumour response to anti-PD-1 and anti-PD-L1 therapy and overall survival [51]. The associations of these radiomic features with gene expression patterns could also be elicited [52]. Table 1 shows a summary of papers that have used radiomics features for risk stratification of patients with NPC at various stages of disease. The imaging type and the number of features extracted varies greatly, and different endpoints have been used to determine prognosis, all with good success. 

### 4.2. EBV-Associated Biomarkers

The pathogenesis of NPC is yet to be fully elucidated, but there is a clear association between Epstein–Barr virus (EBV) infection and the development of NPC [55]. Primary infection of EBV first occurs when the virus enters the oropharyngeal mucosal epithelium and spreads to the nasopharyngeal epithelial cells. While the lytic phase of the infection may have some involvement in tumorigenesis, the latent phase of infection of the nasopharyngeal epithelial cells by EBV, expression of latent EBV genes and subsequent accumulation of genetic mutations is postulated to be the driving force for NPC development [25,56]. 

It has been confirmed that free EBV DNA in peripheral blood arises from tumour cells, and the levels are reliable even for cases of recurrence and metastasis [57]. 

The use of circulating EBV DNA levels for the diagnosis of NPC, monitoring of disease progression, and recurrence post-treatment has been established in practice [58]. A meta-analysis of a large cohort of 27,235 NPC patients in an endemic region found that high EBV DNA levels across various time points of the disease all significantly correlated with poor outcomes, with at least a 2.5-fold increased risk of death [59]. A retrospective study found that changes in EBV DNA levels between different time points also hold predictive value for prognosis. Persistently detectable EBV DNA at two-time points or more had the highest risk for disease progression, distant metastasis, locoregional recurrence and death, and persistently undetectable EBV DNA at the first four time points had the lowest risk [60]. Plasma EBV DNA levels in NPC patients correspond to disease activity [61] and tumour burden, which drop sharply upon surgical resection of the tumour [62]. Undetectable EBV DNA levels 6 months after IMRT effectively predicted longer 3-year survival endpoints (local failure-free, regional failure-free, distant metastasis-free, progression-free, cancer-specific and overall survival) in patients with non-metastatic NPC compared to positive EBV DNA levels at the same time point [63]. 

EBV DNA levels can complement the AJCC/UICC staging system, corresponding well to the T classification [64], and studies incorporating pre-treatment plasma EBV DNA into the eighth edition AJCC/UICC TNM staging, as well as other prognostic nomogram models, have shown better prediction of survival outcomes compared to using the TNM classification alone [65,66,67,68,69]. 

Studies have harnessed EBV DNA as a tool to stratify patients for further treatment. A randomised controlled trial has attempted to use post-treatment EBV DNA levels to guide the selection of patients for further adjuvant chemotherapy after RT. There was no improvement in the 5-year disease-free survival rate from the additional therapy. This was postulated to be related to the additional toxicities, and post-RT EBV DNA levels was still a significant factor for poorer outcomes including locoregional failure, distant metastasis and death, even more so, in fact, than the AJCC/UICC TNM staging [70]. A four-grade systematic risk stratification model has been proposed, using EBV DNA levels to stratify patients into groups for individualised treatment strategies [71]. In a study by Q. Zhang et al., the number of cycles of induction chemotherapy was adjusted based on the EBV DNA load post-induction chemotherapy, as well as the TNM stage. While survival outcomes remained similar with the new treatment regime, toxicity was reduced [72]. Twu et al. showed in a retrospective study that patients with persistently detectable plasma EBV DNA after RT yielded reduced distant relapse and improved OS with the addition of adjuvant oral tegafur-uracil for one year [73]. Chan et al. have attempted to use EBV DNA levels to select patients with detectable post-IMRT EBV DNA levels for an additional six cycles of adjuvant chemotherapy, but it has not been shown to benefit relapse-free survival. Capecitabine maintenance therapy was shown to improve overall survival (OS) in dmNPC for patients with pre-treatment EBV DNA titres of ≤30,000 copies/mL [74]. A high pre-operative EBV DNA level prior to salvage surgery may identify patients with recurrent NPC that are at a higher risk of subsequent distant failure [75].

The measurement of circulating EBV DNA levels using RT PCR has the potential to be a routine clinical investigation in view of its speed, accuracy, and ability to be carried out in large volumes. However, the lack of harmonisation of the assay for the measurement of EBV DNA levels across laboratories worldwide has led to a lack of consistency and difficulty in the comparison of results across different studies [76]. The optimal cut-off EBV DNA levels for risk segregation and prognostication varied greatly among different studies, with certain studies using recursive partitioning analysis to obtain optimal cut-off levels for analysis [77]. Further studies to determine an optimal EBV DNA quantity to segregate patients of the same stage into different outcomes would be beneficial.

The use of EBV DNA to risk stratify patients in non-endemic areas may be more challenging, as a larger proportion have histological subtypes that do have such a strong association with the EBV virus. While studies have shown the association of EBV DNA levels of NPC patients from non-endemic regions with both clinical and TNM stages [78], the application of this biomarker for diagnosis and prognostication is limited by lower sensitivity compared to endemic regions [79]. 

Other biomarkers that are linked to EBV include IgA antibodies for early antigen (EA), viral capsid antigen (VCA), and nuclear antigen (EBNA1). During the EBV proliferation cycle, viral antigens are expressed, triggering the body to produce these antibodies. EBV-based antibodies have already been incorporated into clinical use for the screening and diagnosis of NPC in high-risk patients, with serum EA-IgA and VCA-IgA being the most frequently used markers and often in combination [80]. 

There has been growing interest in whether these diagnostic markers could assist with prognostication as well, complementing the usage of EBV DNA levels. Existing studies have conflicting findings regarding this. A study by P Sun et al. found both EA-IgA and VCA-IgA to correlate positively with the T and N classification and the disease stage of the seventh edition of the AJCC/UICC NPC staging system [81]. A meta-analysis found that EA-IgA was an important prognostic indicator in patients with NPC but not VCA-IgA [82]. Another study, which only included patients with undetectable pre-treatment EBV DNA levels in order to reduce the inference of EBV DNA levels on prognosis, showed that both EA-IgA and VCA-IgA did not have prognostic value for patients with NPC [83]. 

A study has compared 6 plasma biomarkers related to EBV. These included two EBV DNA fragments (BamHI-W 76 bp and EBNA1 99 bp) and four anti-EBV antibodies (EA IgA, EA IgG, VCA IgA, EBNA-1 IgA). Only EBNA1 99 bp has been shown to have prognostic value in this study [81]. In a study conducted in a non-endemic region of Russia, VCA-IgA antibodies did not correlate with the patient’s clinical status and prognosis, while EBV DNA levels only corresponded to the “N” staging of the disease [84]. When it comes to prognostication, EBV DNA levels ultimately appear to have stronger clinical utility compared to levels of EBV serological markers. The difference is possibly due to the fact that IgA antibodies are produced by an immune response to the proliferation of EBV, which can vary greatly as a result of differing immunity and the possibility of immunodeficiency diseases, while plasma EBV DNA arises from the apoptosis of cancer cells, which occurs in active disease states [84]. 

### 4.3. Histological Subtypes Associated with Prognosis

The histological classification of NPC has evolved over time. Currently, the World Health Organisation (WHO) has divided it into three subtypes: keratinising squamous cell carcinoma (KSCC), non-keratinising carcinoma (NKC), and basaloid squamous cell carcinoma (BSCC). NKC tumours are further categorised into undifferentiated carcinoma (NKUC) and differentiated carcinoma (NKDC) [85,86]. The NKUC subtype has two distinct patterns, the Regaud and Schminke patterns [87]. The Regaud pattern shows cohesive cells with clear cell margins differentiating from tumour epithelial and microenvironment, while the Schminke pattern has non-cohesive cells with a mixture of infiltrating immune cells into the tumour compartment, as shown in Figure 1.

Currently, the histopathological subtype of NPC is not included in the AJCC/UICC staging due to its supposed lack of clinical significance. For instance, there is little difference in prognosis between undifferentiated and differentiated non-keratinising carcinomas, which account for most NPC cases in China [86,88]. However, after an extensive follow-up, Wu et al. stated that NKDC has a poorer specific survival compared to NKUC during a follow-up period of 5 years, although this number becomes comparable after more than 5 years of follow-up [89]. 

There have been studies proposing a new pathologic classification, which could provide insights into its prognostic value. For instance, Wang et al. classified NPC into four subtypes: epithelial carcinoma (EC), sarcomatoid carcinoma (SC), mixed sarcomatoid-epithelial carcinoma (MSEC), and squamous cell carcinoma (SCC). They identified each subtype’s overall survival and responsiveness towards CCRT [90]. The results showed that, compared to RT alone, CCRT is more effective in treating advanced patients with EC and MSEC subtypes. However, the SC and SCC subtypes were observed to be comparatively less sensitive to chemoradiation; hence, patients might need more aggressive therapies like high-dose irradiation and/or adjuvant chemotherapy. 

Another intriguing point about NPC tumour histology is the correlation between intratumoural and stromal tumour-infiltrating lymphocytes (TILs) and disease prognosis [91]. NPC tumours are characterised by heavy non-malignant lymphocyte infiltrates and are often termed “lymphoepitheliomas” [92]. The NPC tumours’ interaction with the surrounding microenvironment, which consists of a plethora of immune cells, have been reviewed in great detail by Liu et al. [93]. 

A study by Almangush et al. in a non-endemic population showed that NPC cases with low intratumoural TILs had poor overall survival and disease-specific survival. The prognosis is even poorer for keratinised tumours with low intratumoural TILs [94]. Similar results have also been obtained in another large cohort study that showed that high TILs are significantly associated with better disease-free survival, overall survival, as well as distant metastasis and local-regional recurrence-free survival [95]. Interestingly, a study by Zhang et al. observed that a high stromal percentage of TILs was significantly associated with better PFS but not intratumoral TILs [96]. The study revealed that stromal % TILs have a negative association with the activation of TGF-β and WNT transcriptional pathways, as well as the mutations in the P13K and MAPK pathways. Various studies were conducted to identify the types of immune cells present and their relation to disease progression and survival. A study by Zhang et al. revealed that there were different subsets of immune cells affecting survival depending on the stages of the disease [97]. Better OS and PFS were associated with higher densities of FOXP3+ TIL or FOXP3+ TIL combined with GRB+ TIL (*p* < 0.01) in all patients and in patients with late-stage diseases (Stages III and IV, *p* < 0.01). On the other hand, for early-stage patients, lower densities of CD8+TIL or a high ratio of FOXP3+TIL to CD8+TIL correlated with better PFS (*p* < 0.05).

In other study cohorts of NPC tumours, PD-L1 expression on tumour cells in combination with a higher CD8+ TIL density was significantly associated with a favourable prognosis, while positive PD-L1 with a low CD8+ TIL density was associated with poor prognosis [98]. This result is concordant with the findings by Ooft et al., who stated that increased CD8 count and PD-L1 co-expression are associated with better OS (HR 0.073 (95% CI 0.010–0.556)) in EBV-positive NPC and NPC group as a whole [99]. In addition, in EBV positive NPC tumour specifically, the co-expression of CD8 and PDL1 showed better DFS (HR 0.407 (95% CI 0.195–0.850)) and OS (HR 0.170 (95% CI 0.037–0.787)).

Even though the results of these studies vary in the combination of biomarkers, overall, TILs have the potential to be a predictive biomarker for NPC risk stratification by predicting the disease progress and responses to immunotherapy. 

## 5. Genomic Biomarkers for Risk Stratification

Unlike other forms of cancers, which have options for tumour-selective therapy with minimum toxicity, NPC treatment still relies on chemotherapy and radiotherapy as the first-line treatment [100,101,102]. Although the initial response to therapy is favourable, disease recurrence and distant metastasis pose a challenge to treatment success [103]. Molecular and genomic-wide studies have been conducted to observe the changes in the NPC molecular landscape and identify key biomarkers and pathways for more targeted therapy.

### 5.1. Genetic Alteration in NPC Molecular Landscape

Genetic alterations in NPC and hypermethylation of DNA have been studied as potential factors in NPC carcinogenesis. For instance, genetic losses and gains were identified in the primary tumour of NPC patients. Genetic loss was detected in chromosomes 1p, 3p, 9p, 9q, 11q, 13q, 14q, and 16q, while gains were found in chromosomes 1q, 8, 12, 19, and 20 [104]. Lo et al. suggested that the deletion in chromosomes 3p, 9p, and 14q was consistent throughout the microdissected NPC samples, and thus, the inactivation of tumour suppressor genes in those chromosomes is critical for NPC to develop. Even though its clinical significance remains elusive, it has been shown that loss of heterozygosity (LOH) in certain chromosomes might be responsible for aggressive and metastatic NPC, as the significantly high frequency of LOH on 9p21 and 19q13 was detected only in the T3 and T4 stages of NPC [105]. Despite the presence of genomic alterations, targeting actionable mutations in NPC remains a challenge. 

The mutational landscape of NPC has been explored to identify a potentially suitable drug target in an affected pathway. Lin et al. conducted whole-exome, targeted deep sequencing, and SNP array analysis of 128 NPC cases to uncover distinct mutational signatures and nine significantly mutated genes [106]. This study highlighted alterations targeting the chromatin modification pathway, which may contribute to the aggressiveness of NPC. ARID1A was one of the most frequently mutated genes and was previously shown to have an inhibitory effect on cell proliferation [107,108]. Furthermore, this study also showed that the ERBB-PI3K signalling pathway has been altered through the mutation of *PIK3CA*, *ERBB2* or *ERBB3,* in which a similar conclusion was attained by another study [109]. This appeared to occur in advanced clinical stage tumours, suggesting that such mutations increase the malignant behaviour of NPC tumours. A recent study by Zhou et al. also found that *PIK3CA* and *SF3B1* mutations were key to distant metastasis. The authors combined genomic studies, tumour mutational burden (TMB), and N stage to generate a more accurate prediction of distant metastasis-free survival (DMFS). Further validation cohort studies will be necessary for future clinical implementation [110]. Unfortunately, a phase II study evaluating the effect of an AKT inhibitor was terminated as the level of clinical activity during the first stage of enrollment (21 patients) did not meet the requirement to advance to the second stage [111]. This study has shown that MK-2206 (AKT inhibitor) treatment had limited clinical effects for a cohort of patients with metastatic NPC who were not selected based on the gene expression of genomic aberration in P13K-AKT pathways. However, if the patients were to be screened based on tumoral expression of *P13KCA* aberration, the enrollment is going to be challenging as based on the whole-exome sequencing study of NPC, the actual prevalence of *PIK3CA* mutation and amplifications was considerably low, at 6%.

Another pathway that was also highlighted to be the core of NPC development is NF-κB [112,113,114]. Li et al. identified genomic aberrations on multiple regulators of the NF-Κb pathway, which affected both canonical and non-canonical pathways [112]. In fact, 41% of NPC cases experienced mutations in various negative regulators. For instance, *TRAF3* was found to be mutated in 8.6% of NPC cases, mostly in the RING finger and the MATH/TRAF domains, which are essential areas for suppression of NIK-activating NF-κB signalling. This gene is a key negative regulator of the non-canonical wing of the NF-κB pathway in NPC. Mutation of *NLRC5* (4.8%) was also observed in the study cohort, albeit not statistically significant. This gene is an inhibitor of NF-κB activation and Type I interferon signalling, which competes with NEMO for binding to IKK-alpha and IKK-beta. Therefore, it is able to block their activation and kinase activities. 

Interestingly, they observed mutual exclusivity among tumours with genomic aberrations and those with overexpression of EBV oncoprotein latent membrane protein 1 (LMP1). LMP1 is an NF-κB signalling activator in NPC, and 25.7% of the cases in the cohort have high LMP expression, which also has been associated with poor outcomes. It seems that the deregulation of this pathway, either through somatic genetic mutations or viral events, is a core event in NPC pathogenesis.

Unfortunately, targeting the NF-κB pathway is challenging. Despite the clear linkage of the NF-κB pathway in cancer pathogenesis, none of the NF-κB inhibitors have been clinically approved. This pathway functions in diverse roles, such as inflammatory responses, immune regulation, cell proliferation, and survival. Therefore, suppressing the NF-κB pathway globally is feared to cause adverse reactions in patients [115].

Despite the abundance of data from the study of whole exomes and genomes, it is still a challenge to process the amount of data required for clinical application in a cost-effective manner for individualised diagnosis and treatment. To reduce costs, there has been a growing interest in targeted next-generation sequencing to test solely the genes that are clinically significant.

### 5.2. Gene Expression Profiling in NPC

Differentially expressed genes (DEGs) between tumours and normal tissues, as well as between tumours, have the potential to be used in precision medicine. When properly curated, gene expression signatures could identify potential therapeutic markers and reflect the ongoing biological processes. Hence, they are beneficial in predicting the risks of recurrence, distant metastasis, and/or treatment response [116,117]. Table 2 summarises the studies that generated NPC gene expression signatures using RNAseq, microarray, and bioinformatics analysis methods. They identified gene sets that could potentially be novel biomarkers, which could stratify patients based on the metastasis risk or the responsiveness to a certain treatment. Those that were classified as a high-risk group generally had poorer prognosis with shorter OS, PFS, and/or DMFS and were less likely to respond to treatment. In addition, some gene signatures could also differentiate NPC from normal samples, opening up the potential for early NPC detection. 

An earlier study conducted in 2012 used peripheral whole blood samples collected from NPC patients and healthy controls (66 NPC; 33 controls). Microarray analysis was used to evaluate the blood-based gene expression signature and identified three genes, *LDLRAP1*, *PHF20*, and *LUC7L3*, that could differentiate NPC from controls and patients affected by other diseases [118]. These genes are known to be tumour-associated antigens and/or to be partaking in cellular signalling processes. In addition, these genes could potentially predict treatment response. For instance, LUC7L3, or cisplatin resistance-associated overexpressed protein (CROP), was found to be significantly lower in NPC samples than in controls and other cancer samples. This gene can potentially stratify patients into responders and non-responders of cisplatin, a widely used NPC treatment.

A study on 60 tumour biopsies using RNA Seq technology identified 13 significant genes between the recurrent/metastatic group and no recurrent/metastatic group [120]. Based on these genes, they then proposed a 4-mRNA signature consisting of U2AF1L5, TMEM265, GLB1L and MLF1. Cell proliferation and immune-related hallmarks such as “G2/M checkpoint” and “interferon gamma response” have been enriched in the gene set enrichment analysis. This study highlighted GLB1L, which potentially plays a key role in governing immune cell recruitment to NPC tumours, as the expression of this gene is positively correlated with immune infiltrating cells. The receiver operating characteristic (ROC) and Kaplan-Meier (K-M) analyses indicated that this gene signature had good prognostic value for NPC. The area under the curve (AUC) values of the signature were higher than those of the T stage and N stage for OS (0.893 vs. 0.619 and 0.582, respectively) and PFS (0.86 vs. 0.538 and 0.622, respectively). Through the K-M curve, it was also shown that there was an increase in risk score in both OS and PFS (OS, HR (95% CI):2.72(1.679–4.400) and PFS, HR (95% CI):1.92(1.446–2.563)) which indicated the poorer outcome of patients with a high-risk score. Both the OS and PFS of the high-risk group are significantly shorter than those in the low-risk group (*p* < 0.05).

A large multicentre cohort study was conducted in China with more than 900 patients with locoregionally advanced NPC [119]. The tissue samples were obtained from paired locoregionally advanced NPC tumours with and without distant metastasis after radical treatment. This study developed a distant metastasis gene signature (DMGN) classifier consisting of 13 genes to classify patients into high-risk and low-risk groups, where patients in the high-risk group were found to have a shorter distant metastasis-free survival (DMFS). Furthermore, the patients from the low-risk group responded to concurrent chemotherapy, unlike the high-risk group.

Immune gene signatures also seem to play a role in improving prognosis. Through a study using an in-silico deconvolution algorithm, CIBERSORT, it is evident that the immune infiltration profile of NPC tumours is significantly different from normal tissue [122]. NPC tumours contain a higher proportion of M1 macrophages while having a lower proportion of memory B cells and CD4 memory resting T cells, which may be associated with tumourigenesis of NPC. In addition, single-cell gene expression studies also show that the infiltration of immune cells into the microenvironment of NPC plays a vital role in the disease progression and prognosis. NPC tumour microenvironment (TME) is highly heterogenous, and it has been shown that immune subtype-specific signatures of macrophages, natural killer (NK) cells, as well as plasmacytoid dendritic cells significantly improve the survival outcome [123,124].

### 5.3. miRNA Studies

miRNA are short, single-stranded, non-coding RNA molecules that are responsible for gene regulation [125]. The dysregulation of miRNA has been evident in numerous cancer types, including NPC. miRNA signatures could potentially stratify patients for risk of survival, metastasis, and treatment response. Table 3 summarises studies generating miRNA signatures from paraffin-embedded tissue and blood samples. Some of these signatures detected NPC and differentiated them from normal tissues or other diseases, while the others focused on the high- and low-risk groups of developing metastasis in a certain period of time. However, from these mentioned studies, there was little consistency in the miRNA presented, even though in general, these miRNAs play a key role in promoting or suppressing tumour formation, such as apoptotic pathways.

Li et al. have studied stage-specific miRNA profiling from NPC tissue, resulting in three miRNA signatures [125]. It was observed that certain miRNAs (miR-203, miR-199b-5p, and miR-4324) were down-regulated in stage I NPC and have been known to play an important role in suppressing apoptosis pathways. In addition, they have also found 49 other dysregulated miRNAs across every disease stage compared to normal nasopharyngeal tissues.

In an earlier study in 2012, a 5-miRNA signature that resulted from miRNA profiling of NPC tissue was developed (miR-142-3p, miR-29c, miR-26a, and miR-30e) by Liu et al. [126]. Using the miRNA signature, patients were stratified into high and low risk. The high-risk patient group had shorter DFS, DMFS, and OS values. The analysis has also shown that patients in advanced stages (III and IV) who have high-risk scores did not benefit from concurrent chemotherapy as much as those with low-risk scores. Overall, it is shown that the combination of TNM staging with this miRNA signature had a better prognostic value than TNM staging alone.

Bruce et al. generated a 4-miRNA signature from NPC tissue consisting of miR-154-5p, miR-449b-5p, miR-140-5Pe, and miR34c-5p [127]. This signature stratified patients into low and high risk. The high-risk group had a higher possibility of distant metastasis and thus would benefit from the administration of combined chemoradiation therapy to reduce the 5-year risk of developing distant metastasis. Bruce et al. also compared this signature with the 4-miRNA signature by Liu et al., with only miR-30e overlapping between these two studies and the 5% most frequently occurring miRNAs from other random signatures, the data found that among the enriched gene sets, cell cycling (“Cell Cycle” KEGG pathway), and proliferation are among the significant biological processes that mediate the distant metastasis underlying these miRNAs.

## 6. Limitations Faced in Clinical Implementation of Predictors

While research has yielded good data on the utility of these predictors, there are still issues facing clinical implementation. There has not been a consensus on the standardisation of the assays used for EBV DNA detection. Similarly, there is no standardised protocol yet to analyse the imaging data and to select radiomic features that give the most significant prognostic value; hence, the consistency of performance varies.

In gene expression profiling, small tumour size and low tumour purity remained key challenges. The size of tissue obtained from nasopharyngeal biopsies is generally limited. NPC tumours are also known to have lower tumour purity, as they are a mixture of immune infiltrating cells, stromal components, and tumour epithelial cells. Thus, lower tumour purity hinders precise genome characterization. Not to mention, these tissue samples usually went through a fixation process and were embedded in paraffin, which further degraded the RNA content in the samples. Advanced technology is therefore needed to extract and sequence these poor-quality RNA. These challenges mainly caused the exorbitant cost of implementing it clinically for individual patients at this point in time.

## 7. Conclusions

Disease recurrence and metastasis seem to be the key challenges in NPC. The current TNM staging has given limited prognostic value to guide treatment strategies and predict treatment response and survival outcomes, and there is a need to enhance this staging system, as summarised in Figure 2. Gene signatures, serological markers, and radiomic features have been explored in this study, and solutions are needed to overcome the limitations and allow practical implementation in the clinical setting.

The factors discussed in this paper can be used in tandem to elicit the intricate differences between NPC patients corresponding to differing prognoses and responses to treatment. Precision therapy can then be applied from the conclusions generated from all this complementary information. Considering the promising developments in the clinical and molecular research of NPC, this could be the future of the staging system of NPC.

## Figures and Tables

**Figure 1 cancers-16-00918-f001:**
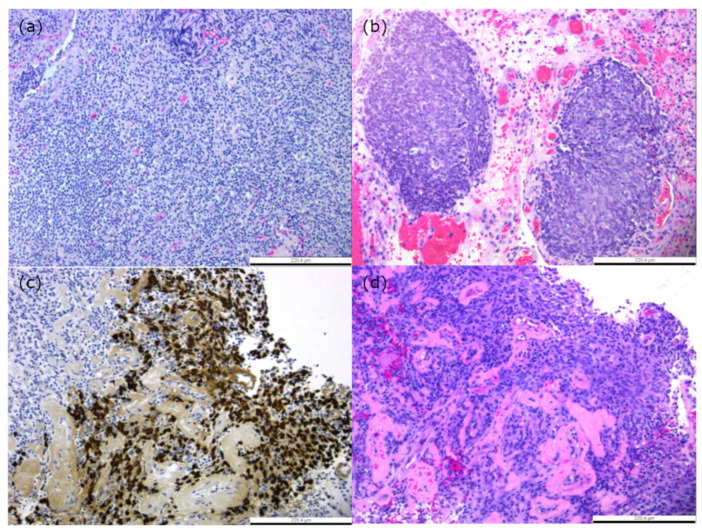
Hematoxylin–eosin staining of NPC tumour sections. (**a**) Schminke pattern. (**b**) Regaud pattern. (**d**) Nuclear staining of EBV-encoded RNAs (EBER) by in situ hybridisation in NPC that is adjacent to the H&E staining in (**c**). Scale bar is 220.4µm.

**Figure 2 cancers-16-00918-f002:**
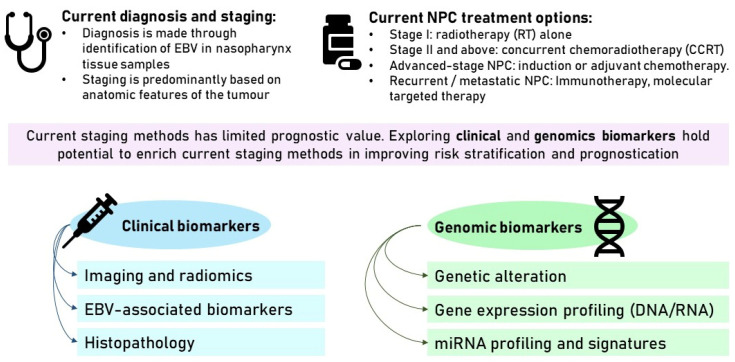
Summary of the topics covered in this review paper.

**Table 1 cancers-16-00918-t001:** NPC stratification based on radiomics features.

Year	Imaging Type	Cohort (Training, Validation)	Number of Radiomic Features Extracted	Endpoint	Reference
2016	^18^F-FDG PET/CT	101 primary NPC patients pre-treatment (101, 0)	15 (4 histogram-based heterogeneity parameters, 6 second-order texture features, 5 higher-order features)	OS, recurrence-free survival (RFS)	Chan et al. [45]
2017	CE-T1WI, T2WI	100 NPC patients pre-treatment (stage III-IVb) (70, 30)	970 (tumour intensity, shape, texture, and wavelet features)	PFS	Ouyang et al. [43]
2019	CE-T1WI, T2WI	277 non-metastatic NPC (217, 60)	525 (11 first-order intensity features, 41 texture features, 5 shape features, with 4 subbands of Coiflet wavelet transforms per MRI sequence)	3-year disease progression	Du et al. [44]
2019	^18^F-FDG PET/CT	707 patients with advanced (Stage III-IVA) NPC (470, 237)	269 (136 deep learning features, 133 handcrafted features—divided into shape, histogram, grey-level co-occurrence matrix, and grey-level run-length matrix features)	Disease-free survival (DFS), distant metastasis-free survival (DMFS), OS, locoregional relapse-free survival (LRRFS)	Peng et al. [47]
2021	CT	311 patients with locally-advanced NPC (stage III-IVa) (218, 93)	1409 (shape, first-order, texture features)	PFS	Yan et al. [53]
2022	CT	197 non-Chinese non-metastatic NPC patients (158, 39)	842 (shape, first-order intensity, texture, and wavelet-based class)	3-year OS, PFS, DMFS	Intarak et al. [46]
2023	CE-T1WI, T1WI, T2WI	329 primary NPC patients pre-treatment (229, 100)	3669 (shape, first-order, grey-level co-occurrence matrix (GLCM), grey-level dependence matrix, grey-level run length matrix, grey-level size zone matrix, neighbouring grey tone difference matrix; 3 directions of wavelet decomposition applied)	Staging classification	Li et al. [54]

**Table 2 cancers-16-00918-t002:** NPC stratification based on gene expression signatures.

Year	Cohort	Methods Used	Sample Source	Gene Expression Signatures	Outcome Measure	Reference
2012	66 NPC patients and 33 healthy controls	Microarray	Peripheral whole blood samples	Primary genes: *LDLRAP1*, *PHF20*, and *LUC7L3* Associated secondary suppressor genes: *EZH1*, *IFI35*, and *UQCRH*	Differentiating NPC from control and other diseases. The average ROC AUC was 0.98 (95% C.I. 0.98–0.99) for the combination of the three primary genes together with their associated suppressor genes.	Zaatar, A.M. et al. [118]
2018	937 patients with locoregionally advanced NPC	Microarray	Paraffin-embedded tissues—patients with stage III–IVa locoregionally advanced nasopharyngeal carcinoma	*YBX3, CBR3, CXCL10, CLASP1, DCTN1, FNDC3B, WSB2, LR1G1, GRM4, ANXA1, WNK1, HDLBP, POLR2M*	Stratifying patients into high risk and low risk; patients with high-risk scores had shorter DMFS (HR 4.93, 95% CI 2.99–8.16; *p* < 0.0001), DFS (HR 3.51, 95% CI 2.43–5.07; *p* < 0.0001), and OS (HR 3.22, 95% CI 2.18–4.76; *p* < 0.0001) patients with low-risk scores.	Tang, X.R., et al. [119]
2021	60 NPC tumour	RNAseq	Paired tumour tissue and normal tissue	*U2AF1L5, TMEM265, GLB1L,* and *MLF1*	Stratifying patients into high risk and low risk; both OS (HR 2.72, 95% CI (1.679–4.400) and PFS (HR 1.92, 95% CI (1.446–2.563) of the patients in the high-risk group were significantly shorter (*p* < 0.05)	Zhao, S., et al. [120]
2022	12 pairs of NPC patients with similar clinical characteristics NPC, but different metastasis risk	RNAseq	NPC biopsy tissue	*AK4*, *CPAMD8*, *DDAH1*, and *CRTR1*	Stratifying patients into high risk and low risk; patients in the high-risk group had a significantly lower DMFS (88.4 versus 73.9%; *p* = 0.00057) and PFS (75.1 versus 60.4%; *p* = 0.0058) than the low-risk group. Low-risk groups could benefit from IC + CCRT, but not those identified as high risk.	Liu et al. [121]

**Table 3 cancers-16-00918-t003:** NPC stratification based on miRNA signatures.

Year	Cohort	Sample Source	MiRNA Signatures	Outcome Measure	References
2012	312 NPC and 18 non-cancer specimens	Paraffin-embedded tissue—primary nasopharyngeal carcinoma and non-cancer nasopharyngitis biopsy specimens	miR-142-3p, miR-29c, miR-26a, and miR-30e	The high-risk patient group had shorter DFS (HR 2.73, 95% CI 1.46–5.11; *p* = 0·0019), DMFS (HR 3.48, 95% CI 1.57–7.75; *p* = 0·0020), and OS (HR 2.48, 95% CI 1.24–4.96; *p* = 0·010). Concurrent chemotherapy did not benefit advanced-stage patients with high-risk scores.	Liu et al. [126]
2015	734 NPC specimens taken from the training and validation cohort	Paraffin-embedded tissue—primary nasopharyngeal carcinoma	miR-154-5p, miR-449b-5p, miR-140-5Pe, and miR34c-5p	High-risk patients had a higher possibility of distant metastasis (DM) (HR 8.25; *p* < 0.001 in the training data set and HR 3.2; *p* = 0.01 in the independent validation set). The high-risk group could benefit from the administration of combined chemoradiation therapy since radiation alone was associated with a 45% risk of developing DM, compared to a 20% risk when treated with combined treatment.	Bruce et al. [127]
2016	Eight patients in stage I–IV and two normal samples	Paraffin-embedded tissue—taken from poorly differentiated squamous NPC patients	Downregulation of miR-203, miR-199b-5p and miR-4324 and upregulation of miR-2117, miR-4502, miR-4494 in stage I NPC	Down-regulation and upregulation of the miRNAs listed might promote the formation of NPC in stage 1 patients. The upregulated miRNAs were found to suppress apoptosis pathways.	Li et al. [125]
2019	120 patients with NPC, 30 patients with head-neck tumours (HNT), and 30 healthy subjects (HS)	Whole blood samples	8 miRNA signature: miR-188-5p, miR-1908, miR-3196, miR-3935, miR-4284, miR-4433-5p, miR-4665-3p, and miR-513b 16 miRNA signature: miR-1224-3p, miR-1280, miR-155-5p, miR-1908, miR-1973, miR-296-5p, miR-361-3p, miR-425-5p, miR-4284, miR-4436b-5p, miR-4439, miR-4665-3p, miR-4706, miR-4740-3p, miR-5091, and miR-513b	The 8-miRNA signature is used to diagnose NPC in training group 1 (96.43% sensitivity and 100% specificity, AUC = 0.995) and validation group 1 (86.11% sensitivity and 88.89% specificity (AUC = 0.941)). 16-miRNA signature is used to differentiate NPC from HNT and HS with 100% accuracy (AUC = 1.000) in training group 2 and 87.04% (AUC = 0.924) in validation group 2.	Wen et al. [128]
2020	200 NPC patients and 189 healthy donors 48 NPC patients and 32 healthy donors	Plasma samples Frozen NPC tissue specimens and paraffin-embedded nasal mucosa tissue specimens from healthy donors	let-7b-5p, miR-140-3p, miR-144-3p, miR-17-5p, miR-20a-5p, miR-20b-5p, and miR-205-5p	None of the seven miRNAs seemed to be associated with NPC prognosis. 7 miRNA signature classified NPC patients from healthy control (AUC 0.807). All of these miRNAs from plasma samples have shown an upregulation trend in NPC compared to healthy patients. Upregulation of miR-144-3p, miR-17-5p, miR-20a-5p, and miR-205-5p and downregulation of let-7b-5p and miR-140-3p in NPC tissue compared to healthy control	Zhang et al. [129]
2022	62 cases of NPC and six cases of non-cancerous tissues	Clinical data set (GSE36682)	ebv-miR-BART19-3p, hsa-miR-135b, and hsa-miR-141	The high-risk patient group has lower OS than the low-risk group (HR 3.98, 95% CI (2.13, 7.42))	Zhou et al. [130]

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
