# Peer review of "Precision Medicine for Nasopharyngeal Cancer—A Review of Current Prognostic Strategies"

_cancers, 2024, doi:10.3390/cancers16050918_

Round 1
Reviewer 1 Report
Comments and Suggestions for Authors
This review gives an overview of current nasopharyngeal carcinoma (NPC) diagnosis, treatment and limitation in the AJCC staging for prognosis, followed by summarising mainly the prognosis test/signatures/models that had been explored based on radiomics, histological subtypes, gene/microRNA signatures etc.
Issues:
1) The title of this review "Precision medicine for nasopharyngeal cancer – a review of current strategies" gives an impression that its content is mainly about stratifying treatment with predictive tests/models. Although section "5.1. Genetic alteration in NPC molecular landscape" seems to match with this title, this review appears to be mainly about prognosis of NPC survival.
2) It was mentioned in line 306 to 308 that further genomic analysis is required for the four new histopathological subtypes but the genomic analysis for these subtypes is actually published already (https://pubmed.ncbi.nlm.nih.gov/34031426/).
3) More recent and/or larger sample size studies/review/meta-analysis about the prognosis of NPC survival based on tumour infiltrating immune cells should be cited and discussed. For examples: https://pubmed.ncbi.nlm.nih.gov/28851814/ and https://anpc.amegroups.org/article/view/6949/html
4) NPC tumour tissues are often heavily infiltrated with immune and stromal cells. The variation in the percentage of cancer cells and percentage of non-cancer cells may affect the reproducibility/validity of prognosis tests that used tumour tissue as input. It would be good if this review can include another column in Table 2 and Table 3 to indicate whether method to control for tumour content (e.g. laser captured micro-dissection, specific threshold for percentage of cancer cells etc) was applied by the cited studies.
Author Response
Dear Reviewer 1,
Thank you for your time to review this manuscript.
- Summary
The authors would like to thank Reviewer 1 for taking the time to review this manuscript. Please find the detailed responses below and the corresponding revisions/corrections in track changes in the re-submitted files.
- Point-by-point response to Comments and Suggestions for Authors
This review gives an overview of current nasopharyngeal carcinoma (NPC) diagnosis, treatment and limitation in the AJCC staging for prognosis, followed by summarising mainly the prognosis test/signatures/models that had been explored based on radiomics, histological subtypes, gene/microRNA signatures etc.
Comments 1: The title of this review "Precision medicine for nasopharyngeal cancer – a review of current strategies" gives an impression that its content is mainly about stratifying treatment with predictive tests/models. Although section "5.1. Genetic alteration in NPC molecular landscape" seems to match with this title, this review appears to be mainly about prognosis of NPC survival.
Response 1: We have changed the title to be more reflective of our review as follows: "Precision medicine for nasopharyngeal cancer – a review of current prognostic strategies."
Comments 2: It was mentioned in line 306 to 308 that further genomic analysis is required for the four new histopathological subtypes but the genomic analysis for these subtypes is actually published already (https://pubmed.ncbi.nlm.nih.gov/34031426/ ).
Response 2: We thank Reviewer 1 for the input, and we have amended the part as follow:
“However, the SC and SCC subtypes were observed to be comparatively less sensitive to chemoradiation, hence patients might need more aggressive therapies like high-dose irradiation and/or adjuvant chemotherapy. However, further genomic analysis is also needed for this model to elucidate the underlying mechanism and molecular basis of the proposed classification model.”
Comments 3: More recent and/or larger sample size studies/review/meta-analysis about the prognosis of NPC survival based on tumour infiltrating immune cells should be cited and discussed. For examples: https://pubmed.ncbi.nlm.nih.gov/28851814/ and https://anpc.amegroups.org/article/view/6949/html
Response 3: We would like to thank Reviewer 1 for the input. These 2 papers have been cited and discussed in ‘Section 4.3 Histological subtypes associated with prognosis’ as follow:
- “…NPC tumours are characterized by heavy non-malignant lymphocyte infiltrates, and are often termed “lymphoepitheliomas” [92]. NPC tumours interaction with the surrounding microenvironment which consists of a plethora of immune cells have been reviewed in great details by Liu et al [93].”
- “Similar results have also been obtained in another large cohort study that showed that high TILs are significantly associated with better disease-free survival, overall survival, as well as distant metastasis and local-regional recurrence-free survival [95]. Interestingly, a study done by Zhang et al. observed that high stromal percentage of TILs were significantly associated with better PFS, but not intratumoral TILs [96]. The study revealed that stromal % TILs have negative association with the activation of TGF- β and WNT transcriptional pathway, as well as the mutations in P13K and MAPK pathway.”
Comments 4: NPC tumour tissues are often heavily infiltrated with immune and stromal cells. The variation in the percentage of cancer cells and percentage of non-cancer cells may affect the reproducibility/validity of prognosis tests that used tumour tissue as input. It would be good if this review can include another column in Table 2 and Table 3 to indicate whether method to control for tumour content (e.g. laser captured micro-dissection, specific threshold for percentage of cancer cells etc) was applied by the cited studies.
Response 4: Upon further reading, most of the studies in Table 2 and 3 did not describe tumor selection step in their methods, except for Bruce et al. in Table 3 which used macro-dissection method to ensure >70% of the materials are malignant epithelial.

Reviewer 2 Report
Comments and Suggestions for Authors
In the review by Luvita Suryani et al., they summarize the current strategies of precision medicine for nasopharyngeal cancer. In general, the manuscript is well-written and informative. However, based on the progress in immunotherapy for nasopharyngeal carcinoma in recent years, I recommend the author provide a new section to characterize these advances in the treatment for nasopharyngeal carcinoma and the associated predictor for the immunotherapy.
Author Response
Dear Reviewer 2,
Thank you for your time to review this manuscript.
Summary
The authors would like to thank Reviewer 2 for the kind comments and suggestions. Please find the detailed responses below and the corresponding revisions/corrections in track changes in the re-submitted files.
Point-by-point response to Comments and Suggestions for Authors
Comments 1: In the review by Luvita Suryani et al., they summarize the current strategies of precision medicine for nasopharyngeal cancer. In general, the manuscript is well-written and informative. However, based on the progress in immunotherapy for nasopharyngeal carcinoma in recent years, I recommend the author provide a new section to characterize these advances in the treatment for nasopharyngeal carcinoma and the associated predictor for the immunotherapy.
Response 1: We thank Reviewer 2 for the kind comments and feedback. We have expanded the section on immunotherapy to include some examples and recent trials. However, most of the evidence on the effectiveness of immunotherapy options is based on recent or ongoing trials, hence the predictors for immunotherapy responses are not established yet. Nevertheless, we have attempted to include some details on that as well.
Please refer to lines 87 to 108 for the additions, thank you.

Reviewer 3 Report
Comments and Suggestions for Authors
This is an interesting review about precision medicine for nasopharyngeal cancer.
The paper is well written. However, some issues remain.
Since tables summarizing the studies are present in the text, the authors should provide information about search strategies (databases, search words).
One or more tables summarizing the main results described in the manuscript may be helpful for the readers.
Author Response
Dear Reviewer 3,
Thank you for your time to review this manuscript.
Summary
The authors would like to thank Reviewer 3 for the kind comments and suggestions. Please find the detailed responses below and the corresponding revisions/corrections in track changes in the re-submitted files.
Point-by-point response to Comments and Suggestions for Authors
This is an interesting review about precision medicine for nasopharyngeal cancer.
The paper is well written. However, some issues remain.
Comments 1: Since tables summarizing the studies are present in the text, the authors should provide information about search strategies (databases, search words).
Response 1: This submission is to be considered a thematic review, not a systematic review. The search was not done according to strict principles of systematic review/meta-analysis.
However, we could share some databases and search words we used as follow:
Table 1:
- Database: Pubmed, ScienceDirect, Web of Sciences
- Keywords: ‘nasopharyngeal cancer’, ’NPC’, stratification’, ‘radiomics’, ‘risk’, prognosis’, ‘nomogram’
Table 2:
- Database: Pubmed, ScienceDirect, Web of Sciences
- Keywords: ‘nasopharyngeal cancer’, ’NPC’, ’stratification’, ’staging’, ’RNA’, ’DNA’, ’genomics’
Table 3:
- Database: Pubmed, ScienceDirect, Web of Sciences
- Keywords: ‘nasopharyngeal cancer’, ’NPC’, ’stratification’, ’staging’, ’miRNA’
Comments 2: One or more tables summarizing the main results described in the manuscript may be helpful for the readers.
Response 2: We agree that having a summary of the main results would be important given the broad topics covered in this review. Therefore, we have added Figure 2 consisting of a summary chart.

Reviewer 4 Report
Comments and Suggestions for Authors
Overall, this is a well-written and enjoyable review. The tables included further provide an easy reading format. According to current information provided, EBV DNA load before and after treatment appeared to be favorable markers for prognosis. Does this correlate to the virus-mediated pathogenesis? Are there studies describing the use of anti-EBV agents in the management of endemic NPC cases? What are the outcome? Could these be discussed in the review?
Author Response
Dear Reviewer 4,
Thank you for your time to review this manuscript.
Summary
The authors would like to thank Reviewer 4 for the kind comments and suggestions. Please find the detailed responses below and the corresponding revisions/corrections in track changes in the re-submitted files.
Point-by-point response to Comments and Suggestions for Authors
Comments 1: Overall, this is a well-written and enjoyable review. The tables included further provide an easy reading format. According to current information provided, EBV DNA load before and after treatment appeared to be favorable markers for prognosis. Does this correlate to the virus-mediated pathogenesis? Are there studies describing the use of anti-EBV agents in the management of endemic NPC cases? What are the outcome? Could these be discussed in the review?
Response 1: Thank you for your insightful comment.
The EBV DNA load does correlate to the virus-mediated pathogenesis, which we have briefly mentioned under “EBV-associated Biomarkers.” We have also included some details and one more reference to make the correlation clearer, please refer to lines 213 and 221.
Anti-EBV agents are indeed an area of interest, particularly in the treatment of recurrent / metastatic NPC, that are not responding to conventional radiotherapy and chemotherapy. We have added information on that under “Current treatment options”. Please refer to lines 87 to 108 for these additions, thank you.
